# Can Urban Sprawl Promote Enterprise Innovation? Evidence from A-Share Listed Companies in China

Zeru Jiang [1], Bo Zhang [1], Chunlai Yuan [1,*], Zhaojie Han [2,*] and Jiangtao Liu [3]

1 School of Government, Peking University, 5 Yiheyuan Road, Hai Dian Qu, Beijing 100871, China; jiangzeru@pku.edu.cn (Z.J.); zhangbo@pku.edu.cn (B.Z.)
2 School of Economics and Management, Harbin Institute of Technology, 92 Xida Street, Nan Gang Qu, Harbin 150001, China
3 Business School, Renmin University of China, 59 Zhongguancun Street, Hai Dian Qu, Beijing 100872, China; liujiangtao@rmbs.ruc.edu.cn
* Correspondence: 2201111159@stu.pku.edu.cn (C.Y.); 22S010006@stu.hit.edu.cn (Z.H.)

**Abstract:** Urban sprawl does not invariably impede factor agglomeration; rather, it can foster poly-centric urban configurations, thereby enhancing productivity and encouraging enterprise innovation. This study investigates the effect of urban sprawl on enterprise innovation using data for A-share listed Chinese companies from 2010 to 2020. The results reveal a significant inverted U-shaped relationship between urban sprawl and enterprise innovation, particularly among large enterprises, well-established entities, non-state-owned enterprises, and those operating in non-manufacturing sectors. Additionally, the effects of urban sprawl on the inverted U-shaped relationship are more pronounced in the north-eastern regions and small cities. Regional integration significantly moderates the inverted U-shaped relationship between urban sprawl and enterprise innovation. This research contributes new insights to the field of enterprise innovation, offering theoretical and empirical support for analyzing the economic implications of urban sprawl.

**Keywords:** urban sprawl; enterprise innovation; regional integration; inverted U-shaped relationship

## 1. Introduction

Urban sprawl is a prevailing phenomenon that accompanies the process of urbanization. As urbanization gains momentum and urban boundaries continue to expand, the growth of the urban population fails to keep pace with the expansion of urban areas, resulting in the emergence of urban sprawl. As a development pattern in urban spatial structure, urban sprawl has certain connections and differences with compact cities and polycentric cities. Tsai (2005) contrasted compact cities with urban sprawl, noting that the former is characterized by concentrated, high-density areas with efficient land use, whereas urban sprawl exhibits lower density and a more dispersed spatial pattern [1]. Mcmillen (2004) and Cladera et al. (2009) studied the rise of subcenters in the polycentric urban development model, which increased the job density in surrounding areas and made the city's employment spatial structure more dispersed, similar to how urban sprawl promotes the outward development of industries [2,3]. Although previous research has extensively studied urban form and density, a unified definition has not yet been established due to the complex impacts of urban sprawl on economic, social, and environmental aspects. In the field of economics, urban sprawl is typically defined as the phenomenon of the disorderly expansion of urban land into suburban areas, characterized primarily by low density, dispersed development, and high dependency on automobiles (Dadashpoor and Shahhossein, 2024) [4].

The economic implications of urban sprawl are intricate and manifest in four key dimensions. First, urban sprawl gives rise to the development of new residential, commercial, and industrial zones, which necessitate a larger workforce and skilled professionals for their

operation. Consequently, urban sprawl can create new employment opportunities. Second, as cities increase in size, their purchasing power and spending capacity also increase (George and Waldfogel, 2003; Chen and Rosenthal, 2008; Lee, 2010) [5–7], fostering business and industrial growth. Third, urban sprawl leads to the expansion of cities, resulting in higher costs associated with city construction and maintenance, including expenditures on infrastructure, transportation facilities, and environmental management (Kakar and Prasad, 2020; Navamuel et al., 2018) [8,9]. Finally, urban sprawl requires additional land for residential, commercial, and industrial purposes, which can lead to an oversupply of land, subsequently driving up land prices (Du, 2011; Glaeser, 2006) [10,11]. This increase in land prices, in turn, has implications for housing and commercial land prices. Consequently, the economic effects of urban sprawl remain uncertain and require further empirical investigation.

This study investigates the economic consequences of urban sprawl. More precisely, we focus on the effect of urban sprawl on enterprise innovation, as innovation is a pivotal engine of economic growth (Batabyal and Beladi, 2016) [12]. Enterprises, as major players in economic activities, play an important role in driving innovation. The innovative pursuits undertaken by enterprises have the potential to drive market transformations, stimulate new demand, and facilitate high-quality economic development. With the advent of urban sprawl, enterprises' production and business activities gradually extend to the periphery of the city, leading to changes in factor allocation and production costs within enterprises, thereby influencing enterprise innovation.

The developmental context of China provides a valuable opportunity to examine the relationship between urban sprawl and enterprise innovation in developing nations. As shown in Figure 1, according to data from the China Statistical Yearbook, there has been a substantial expansion of China's built-up area, growing from 455,658 km$^2$ to 624,020 km$^2$ between 2012 and 2021, representing a remarkable increase of approximately 37%. Concurrently, the urban population surged from 721,750,000 to 914,250,000 within the same time frame—a growth rate of about 26.7%. This indicates that over the past decade, the rate of urban sprawl in Chinese cities has exceeded the rate of population growth, resulting in a significant phenomenon of urban sprawl. China has also made rapid strides in innovation, positioning itself as one of the world's leading innovative countries. The country has made noteworthy achievements in cutting-edge technology domains such as artificial intelligence, quantum communications, and 5G. This leads us to ponder the following question: to what extent does urban sprawl affect enterprise innovation? Addressing this question has critical implications for urbanization research, particularly in developing countries, for two main reasons. First, compared with developed nations, developing countries usually have higher population growth rates and more pronounced urbanization trends. Second, developing countries need to continually enhance their innovation ecosystems, improve the quality and efficiency of innovation, and reinforce the protection of intellectual property rights to propel innovation development to new heights.

Given the above considerations, this study empirically examines the influence of urban sprawl on enterprise innovation using data from China's A-share listed companies, spanning the period 2010–2020. Identifying a causal relationship between urban sprawl and enterprise innovation faces potential endogeneity issues. For instance, urban sprawl alters factor allocation, which can affect enterprise innovation. Conversely, more innovative and dynamic enterprises may choose to establish industrial parks or subsidiaries on the outskirts of the city, thereby influencing urban sprawl. To address endogeneity concerns, we employ instrumental variables in the form of cross-multiplier terms between terrain undulation and the rebar price index (Curci, 2015) and cross-multiplier terms between slope and aluminum price to identify the causal relationship between urban sprawl and enterprise innovation [13]. The empirical results reveal several key findings. First, we observe a significant inverted U-shaped relationship between urban sprawl and enterprise innovation, with regional integration playing a positive moderating role. This suggests that moderate urban sprawl can facilitate enterprise innovation, while excessive urban sprawl

can impede enterprise innovation. Specifically, the relationship between urban sprawl and enterprise innovation exhibits an inverted U-shape in most years, with a particularly pronounced effect observed in 2015, potentially attributable to the implementation of the 'removing counties and establishing districts' policy of 2014. Second, the effect of urban sprawl on enterprise innovation shows heterogeneity across various dimensions. The effect is more pronounced in the north-east regions and small cities, which may be linked to lower levels of economic development in those areas. Moreover, the effect is more pronounced for large enterprises, well-established entities, non-state-owned enterprises, and those operating in non-manufacturing sectors, which may be positively associated with enterprise capabilities.

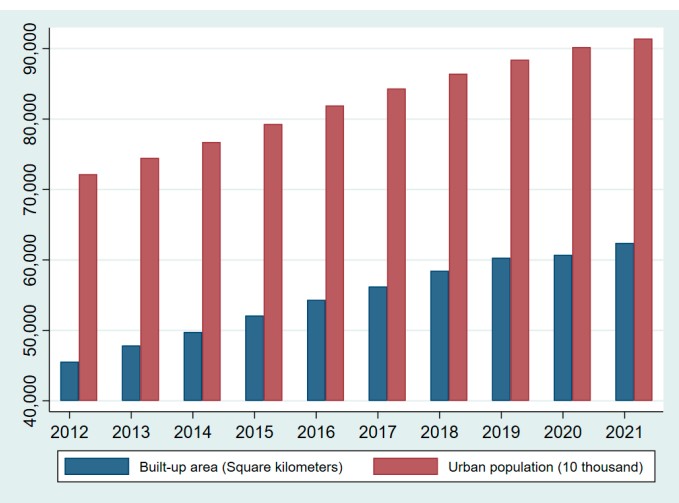

**Figure 1.** China's built-up area and urbanized population, 2012–2021. Source: China Statistical Yearbook, 2013–2022 (http://www.stats.gov.cn/sj/ndsj/) (accessed on 13 April 2024).

The marginal contributions of this article are as follows: First, it enriches the research related to urban sprawl and innovation. Previous studies mainly examined the impact of urban compactness, urban expansion, and agglomeration on firm productivity and innovation. A compact urban form provides richer infrastructure and convenient public transportation, which are more attractive to knowledge-based and innovative talents, thereby promoting regional innovation capabilities (Hamidi et al., 2019; Hamidi and Zandiatashbar, 2018) [14,15]. From the perspective of social interaction, Brueckner and Largey (2008) and Leyden (2003) studied how compact cities, by providing higher-quality urban social life and enhancing social interaction and network usage, increase trust and collaboration among educated millennials, thus fostering innovation activities [16,17]. Tang et al. (2021) empirically analyzed the positive effects of economic agglomeration on urban innovation from the perspective of high-speed rail openings [18]. The opening of high-speed rails accelerates the flow and spillover of innovative elements such as talent and information, improving the efficiency of innovation resource allocation and thereby enhancing urban innovation (Agrawal et al., 2014) [19]. Unlike previous studies that indirectly examined the relationship between urban sprawl and innovation from other aspects of urban spatial structure, this paper directly provides empirical evidence for the impact of urban sprawl on economic activities, particularly corporate innovation, enriching the research at the micro level, i.e., the enterprise level.

Second, it breaks through the linear relationship research between urban sprawl and corporate innovation. The traditional research framework assumes a linear relationship between urban sprawl and innovation, with previous studies analyzing this linear relationship from the perspectives of economic agglomeration, compact urban form, and urbanization. Economic agglomeration strengthens regional economic connections, and a compact urban form facilitates the flow of factor resources, both of which have a positive

impact on urban innovation (Tang et al., 2021; Hamidi et al., 2019) [14,18]. Urbanization, by exploiting economies of scale and agglomeration effects, can enhance total factor productivity and also has a positive impact on innovation (Kumar and Kober, 2012) [20]. Unlike previous studies, we transcend this linear research framework and propose that there is an inverted U-shaped relationship between urban sprawl and corporate innovation, thereby broadening the discussion beyond existing linear conclusions. Moreover, different from studying the inverted U-shaped relationship between urbanization and innovation, which aims to explore the phenomenon of population concentration in urban areas and the resulting economic activities, this paper focuses on examining the impact of urban sprawl on economic activities, particularly innovative activities. This allows us to reconcile two opposing views and provide new insights for enhancing corporate innovation capabilities.

Third, we consider the real factor of regional integration and incorporate it into the framework of the relationship between urban sprawl and corporate innovation, thereby enriching the related research on regional integration. Liu et al. (2023), based on the study of regional integration in China's Yangtze River Delta, show that regional integration can significantly promote innovation in enterprises in the region and that regional integration and the innovation-driven strategy are highly synergistic [21]. Therefore, studying the moderating role of regional integration in the impact of urban sprawl on corporate innovation has significant practical significance for formulating corporate innovation strategies and promoting economic development. Additionally, this paper conducts heterogeneous analyses from multiple perspectives, which can provide stronger recommendations for different cities and types of enterprises to achieve optimized adjustments. We hope our research results can provide theoretical guidance for corporate managers making innovation decisions and offer some suggestions for enhancing corporate innovation performance and innovation vitality.

## 2. Literature Review

Researchers have extensively investigated enterprise innovation. The measurement of enterprise innovation includes innovation inputs, outputs, and efficiency (He and Wintoki, 2016; Wan et al., 2022) [22,23]. The determinants of enterprise innovation predominantly revolve around enterprise characteristics (Shefer et al., 2005; Zhang et al., 2020; Ang et al., 2022) [24–26], market environment (Aghion et al., 2005; Benfratello et al., 2008; Brown et al., 2012) [27–29], institutional environment and government support (Dai and Chapman, 2022; Ding et al., 2023; Su et al., 2022; Xu et al., 2023) [30–33], and urban spatial structure (Lin et al., 2011; Brulhart and Sbergami, 2009) [34,35]. Among the myriad factors that influence enterprise innovation, urban form emerges as a particularly relevant factor in our study.

Regarding urban form's effect on enterprise innovation, the existing literature predominantly investigates the influence of compactness, sprawl, concentration, and fragmentation. Among these factors, compactness, sprawl, and concentration are generally regarded as having a positive effect on innovation. Compact cities tend to attract highly skilled labor (Glaeser and Resseger, 2010) [36], enhancing productivity for both workers and enterprises (Duranton and Puga, 2020) [37]. Benefiting from population concentration and economies of scale, scale expansion promotes innovation and efficiency gains in cities (Fragkias et al., 2013) [38]. The geospatial agglomeration of enterprises fosters knowledge exchange among backward and forward associates, facilitating total-factor productivity spillovers (Baldwin and Okubo, 2006) [39]. Additionally, industrial agglomeration provides more resource advantages and better institutional environments, thereby stimulating enterprises' innovative activities (Panne, 2004) [40]. In contrast, the fragmentation of urban form is characterized by the spatial decentralization of the city, which is often associated with a negative effect on innovation. The spatial fragmentation of residential, commercial, and recreational activities diminishes the effectiveness of land use and hampers overall operational efficiency within the city (Rotem-Mindali et al., 2012) [41]. Furthermore, spatial fragmentation in built-up areas adversely affects the sustainability of coastal cities (Dewa et al., 2022) [42].

Although researchers have extensively studied the potential link between urban sprawl and enterprise innovation, consistent conclusions have yet to be drawn. Specifically, three aspects of the existing literature can be summarized. First, a negative correlation has been observed between urban sprawl and enterprise innovation. As an alternative urban form of urban sprawl, compact cities can have scale effects and agglomeration effects, which are conducive to inter-enterprise factor mobility and improve enterprise productivity (Duranton and Puga, 2020) [37]. Second, a positive correlation has been identified between urban sprawl and enterprise innovation. Over-agglomeration can lead to a crowding effect that diminishes enterprise production efficiency (Lin et al., 2011) [34]. However, the poly-centricity resulting from urban sprawl can harness the agglomeration effect and mitigate the adverse effect of crowding (Huang et al., 2017) [43]. Moreover, excessively densely populated cities tend to have lower levels of interpersonal interactions and social trust (Mouratidis and Poortinga, 2020), hindering the exchange of knowledge and experience and thereby affecting the innovation environment [44]. Finally, as opposed to urban sprawl, a study of China's textile industry found an inverted U-shaped relationship between agglomeration size and labor productivity (Lin, 2011) [34]. Similarly, other studies have revealed nonlinear associations between agglomeration and labor productivity (Brulhart and Sbergami, 2009) [35].

In summary, previous studies have predominantly focused on the relationship between urban form and enterprise innovation or enterprise productivity but have overlooked the direct influence of urban sprawl on enterprise innovation. A study closely aligned with our research is the one by Sun and Li (2022), which delved into industrial agglomeration and regional innovation [45]. However, there is still room for expansion and further exploration. First, the literature predominantly investigates the relationship between the agglomeration of productive services and regional innovation, neglecting nonlinear associations. Second, the literature has mainly examined innovation through the lens of agglomeration, thereby failing to consider the effect of urban sprawl on enterprise innovation. Based on these considerations, this study examines the nonlinear relationship between urban sprawl and enterprise innovation, contributing novel insights to existing research perspectives and providing theoretical references and policy recommendations for enterprise innovation.

## 3. Theoretical Framework

### 3.1. Urban Sprawl and Enterprise Innovation

Urban sprawl has a profound effect on the spatial configuration of cities and the degree of factor agglomeration, which in turn have important implications for enterprise innovation. Moderate urban sprawl can effectively alleviate the adverse effects of excessive agglomeration and foster enterprise innovation. First, excessive agglomeration can result in significant product homogenization among enterprises, leading to decreased factor utilization rates and production efficiency, ultimately diminishing incentives for enterprise innovation. Second, over-agglomeration negatively affects enterprises' export performance (Broersma and Oosterhaven, 2009), which may affect their operating profits and crowd out research and development investment [46]. Furthermore, excessive agglomeration may trigger an 'over-spillover' of technology, talent, and other resources, which can have adverse effects on spillover enterprises. While technological innovation generates positive externalities, small enterprises may lack the capacity to generate original technological advancements and may resort to imitation-based innovation, potentially encroaching upon the interests of larger or more innovative enterprises, thereby inhibiting their motivation to innovate.

However, excessive urban sprawl may also lead to inefficient resource allocation, thereby impeding enterprise innovation. In the context of urban planning in China, local governments possess land development rights (Wang et al., 2020), and urban sprawl has resulted in the conversion of substantial amounts of arable land into urban construction land [47]. To attract investment, the government often provides industrial land at low prices, leading to the inefficient utilization of numerous industrial parks (Du and Peiser, 2014)

and creating a mismatch in land resources [48]. This mismatch extends to transportation resources as well. As urban areas continue to expand, road networks and transportation facilities fail to adequately meet transportation demand, thereby increasing transportation costs and even accelerating energy consumption. These inefficiencies in resource allocation have the potential to crowd out enterprise investment in research and development.

The preceding analysis highlights the existence of an optimal threshold for urban sprawl, beyond which enterprise innovation no longer improves and instead begins to decline. Therefore, the relationship between urban sprawl and enterprise innovation exhibits a nonlinear, inverted U-shaped pattern (Figure 2). Based on this, we propose

**Hypothesis 1.** *There is an inverted U-shaped relationship between urban sprawl and enterprise innovation.*

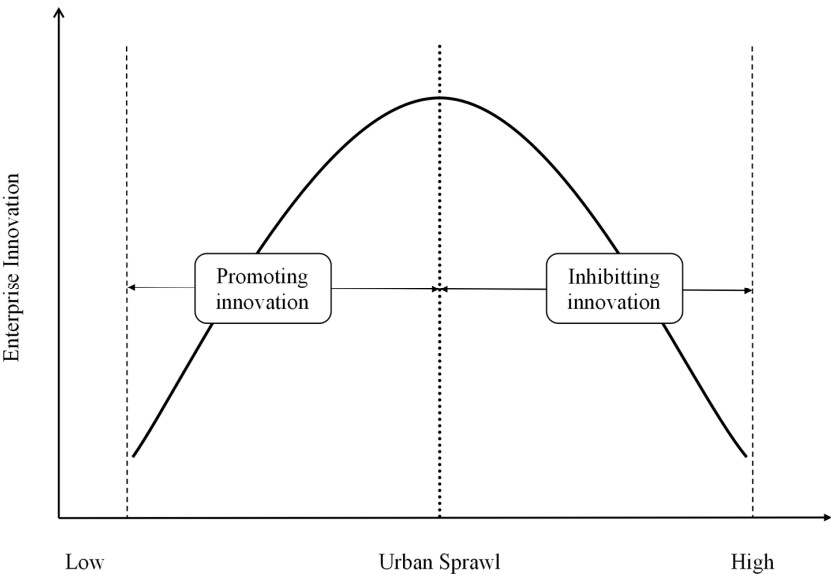

**Figure 2.** Effect of urban sprawl on enterprise innovation.

### 3.2. Urban Sprawl, Regional Integration, and Enterprise Innovation

In most cities around the world, land development is regulated by governments, and local government behavior is largely influenced by national policies and related decisions (Duranon and Puga, 2015; Gyourko and Molloy, 2015) [37,49]. For example, after facing a fiscal crisis in the 1990s, the Chinese central government introduced a tax-sharing reform that effectively addressed the issue of insufficient central fiscal resources but also led to a mismatch between local fiscal authority and responsibilities. Specifically, local governments were left with lower fiscal revenues yet higher fiscal expenditures, and were also responsible for promoting local economic growth and improving public service provision. Consequently, after the decentralization of land control rights through the 1998 Land Administration Law, local governments gradually acquired the rights to manage and trade land resources (Han and Kung, 2015; Chen and Kung, 2016), and eventually developed land finance models such as "low-cost industrial land transfer" and "land sales for fiscal revenue" [50,51]. Although the low-cost land transaction model plays an important role in attracting investment and boosting economic growth (Wang et al., 2024), it also leads to the extensive and inefficient use of industrial land, and even triggers excessive land expansion [52]. Thus, phenomena like urban sprawl and urban expansion are largely caused by local government land transactions. Additionally, since land transactions benefit local fiscal revenues and official promotions, promotion incentives often lead local governments to implement protective measures to safeguard local economies and business development, resulting in increased factor flow and trade barriers between cities within

a province, and varying degrees of regional integration. Therefore, in the process where urban sprawl impacts corporate innovation, regional integration plays a moderating role.

In the left segment of the inverted U-shaped curve, urban sprawl facilitates enterprise innovation by mitigating inefficiencies in factor utilization, productivity, and innovation spillovers arising from excessive agglomeration. Under moderate urban sprawl, higher regional integration ensures that there are fewer territorial barriers between regions, which not only facilitates the sharing of resources and exchange and cooperation in research and development, innovation, and technology transformation among enterprises, but also leverages the advantages of intra-regional factor agglomeration and improves the efficiency of resource utilization by enterprises, thus facilitating enterprise innovation. In contrast, the right segment of the inverted U-shaped curve illustrates how urban sprawl inhibits enterprise innovation through resource mismatches and the crowding out of research and development investment. Under excessive urban sprawl, higher regional integration increases intra-regional transportation costs and decreases resource allocation efficiency. The escalation in production costs, including transportation, not only affects enterprises' operating profits but also crowds out research and development investment, thereby impeding enterprise innovation. Based on this, we propose

**Hypothesis 2.** *Regional integration has a positive moderating effect on the inverted U-shaped relationship between urban sprawl and enterprise innovation.*

## 4. Methods and Data

### 4.1. Model

The benchmark regression model is as follows:

$$\ln Patent_{ijt} = \alpha_0 + \alpha_1 sprawl_{jt} + \alpha_2 sprawl_{jt}^2 + \sum \alpha_3 control_{ijt} + \sum \alpha_4 control_{jt} + \gamma_{year} + \rho_{pro} + \lambda_{ind} + \mu_{ijt}$$

where $i$, $j$, and $t$ denote enterprise, city, and year, respectively. The dependent variable $\ln Patent_{ijt}$ is the innovation level of enterprise $i$ in city $j$ in year $t$. The independent variable $sprawl_{jt}$ is the urban sprawl index of city $j$ in year $t$, and $sprawl_{jt}^2$ is the square of the urban sprawl index. $control_{ijt}$ refers to a series of enterprise-level control variables affecting the innovation of enterprises, while $control_{jt}$ refers to a series of city-level control variables. $\gamma_{year}$ is the year-fixed effect, $\rho_{pro}$ is the province-fixed effect, and $\lambda_{ind}$ is the industry-fixed effect. $\mu_{ijt}$ is the random error term.

### 4.2. Main Variables

#### 4.2.1. Dependent Variables

The existing literature mainly assesses enterprise innovation using indicators such as research and development investment (He and Wintoki, 2016), the number of granted patents, and the number of patent applications filed [22]. However, accurately measuring enterprise innovation poses challenges owing to variations in research and development investment measurements across enterprises and the complex, time-consuming procedures involved in patent grant approvals. In contrast, patent applications can provide a more timely reflection of the outcomes of research and development investment. Therefore, we adopt the approach employed by Xu et al. (2023), which measures enterprise innovation by taking the logarithm of the number of patent applications in the current year plus one [33].

#### 4.2.2. Independent Variables

Currently, there are three main approaches used to measure urban sprawl. The first involves the use of single-indicator measures, such as population density, employment density, and residential density (Kahn, 2001; Lopez and Hynes, 2003) [53,54]. The second approach entails employing a multi-indicator measurement method that utilizes techniques such as principal component analysis to synthesize various indicators into a comprehensive

indicator, including density, land use, and urban form (Wang et al., 2020; Galster et al., 2001; Hamidi et al., 2002; Song and Knaap, 2004) [55–58]. Finally, in the context of rapid advancements in satellite communication technology and remote sensing, the third approach involves leveraging night-time lighting data to assess urban sprawl (Fallah et al., 2011) [59].

The key independent variable in this study is urban sprawl, which is quantified using the urban sprawl index developed by Fallah et al. (2011) and measured using LandScan global population data [59]:

$$SA_i = 0.5 \times (LA_i - HA_i) + 0.5$$

$$SP_i = 0.5 \times (LP_i - HP_i) + 0.5$$

where $SA_i$ is the urban sprawl index considering urban land, $LA_i$ is the proportion of land area in each urban region where the population density is lower than the national average density, and $HA_i$ is the proportion of land area within each urban region where the population density is higher than the national average density. Since the sum of $LA_i$ and $HA_i$ is 1, $SA_i$ can be calculated. Likewise, we can obtain the sprawl index considering population density, denoted as $SP_i$. The urban sprawl index, which combines both dimensions, can be derived as follows:

$$Sprawl_i = \sqrt{SP_i \times SA_i}$$

To gain further insights into dynamic changes in urban sprawl in each region, we use ArcGIS 10.8 to depict the urban sprawl index, as shown in Figures 3–6. Overall, the western side of the Heihe–Tengchong Line exhibits a higher level of urban sprawl, which can be attributed to the relatively lower population density in that region. Over time, the northeastern region shows a gradual decline in urban sprawl, likely influenced by a significant decrease in population and subsequently a reduced demand for urban expansion. In contrast, the eastern coastal region shows a progressive increase in urban sprawl. Due to robust economic development, the eastern coastal regions can attract more populations and new businesses, necessitating the expansion of urban boundaries to accommodate the growing industrial and housing needs.

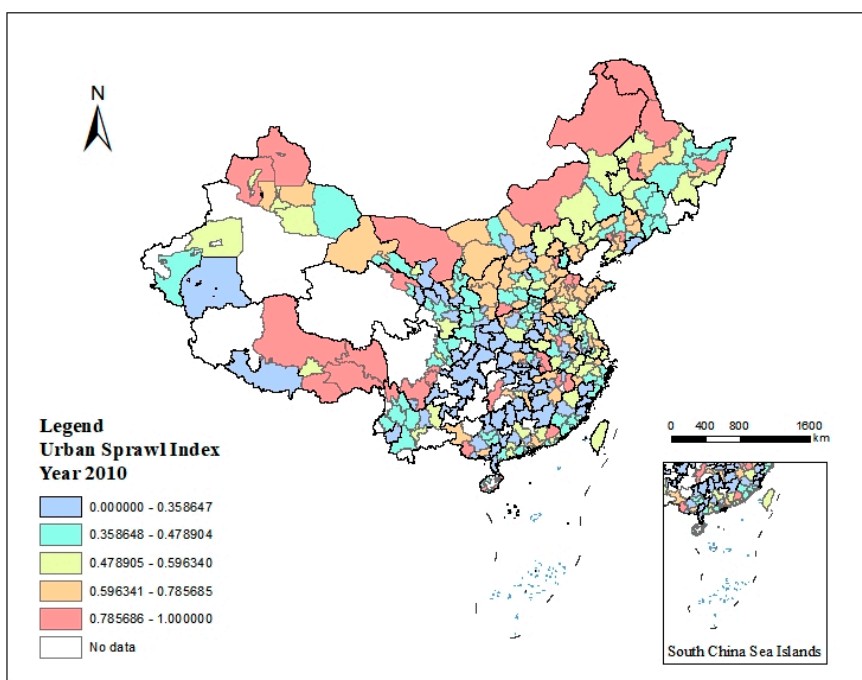

**Figure 3.** Distribution of urban sprawl by province in China, 2010. Source: LandScan global population data, 2010.

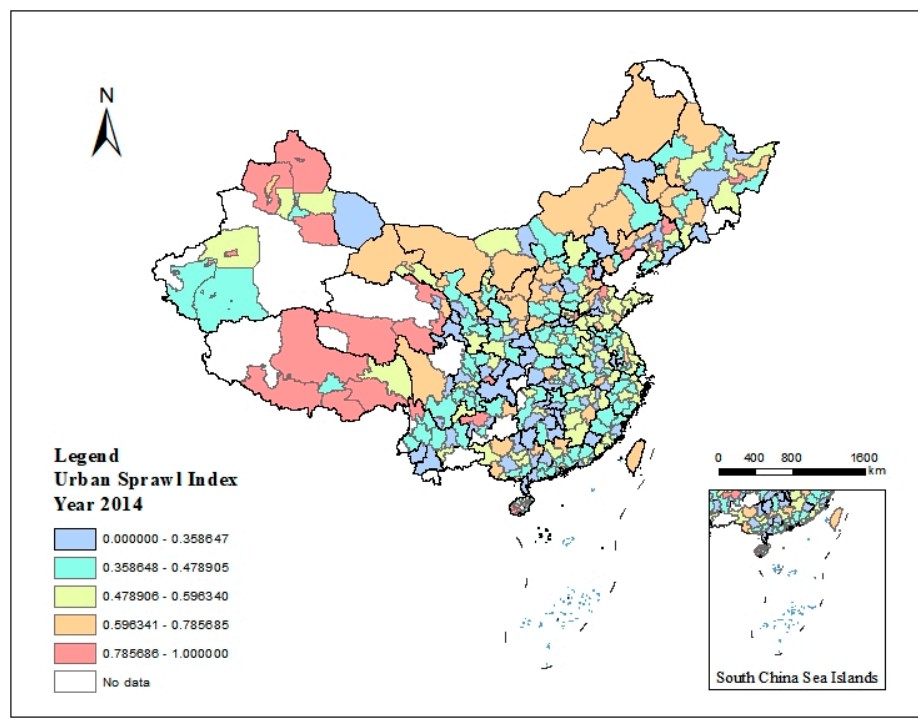

**Figure 4.** Distribution of urban sprawl by province in China, 2014. Source: LandScan global population data, 2014.

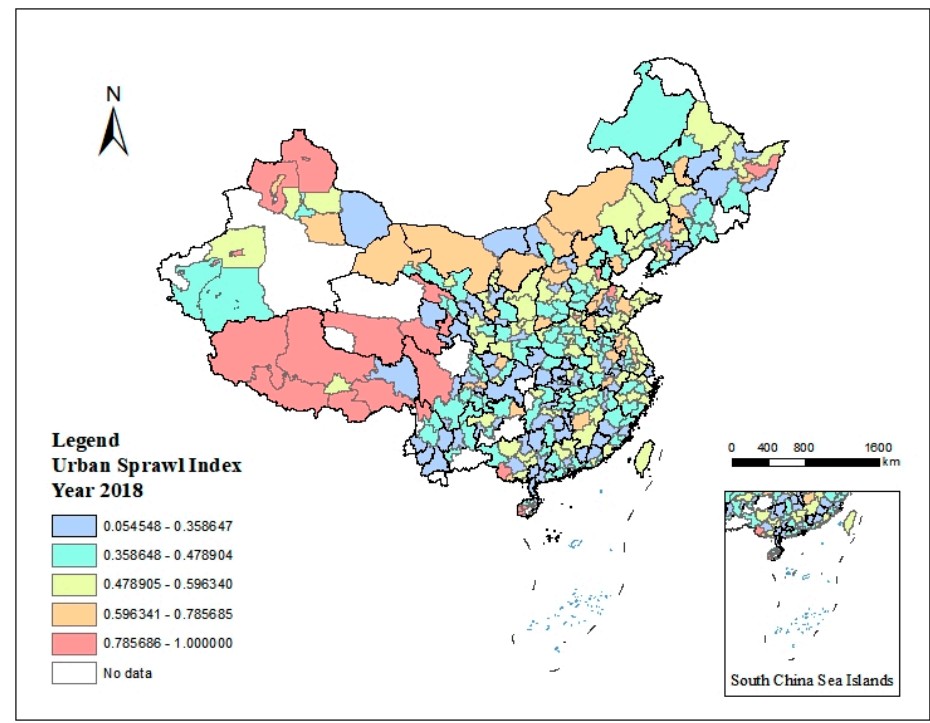

**Figure 5.** Distribution of urban sprawl by province in China, 2018. Source: LandScan global population data, 2018.

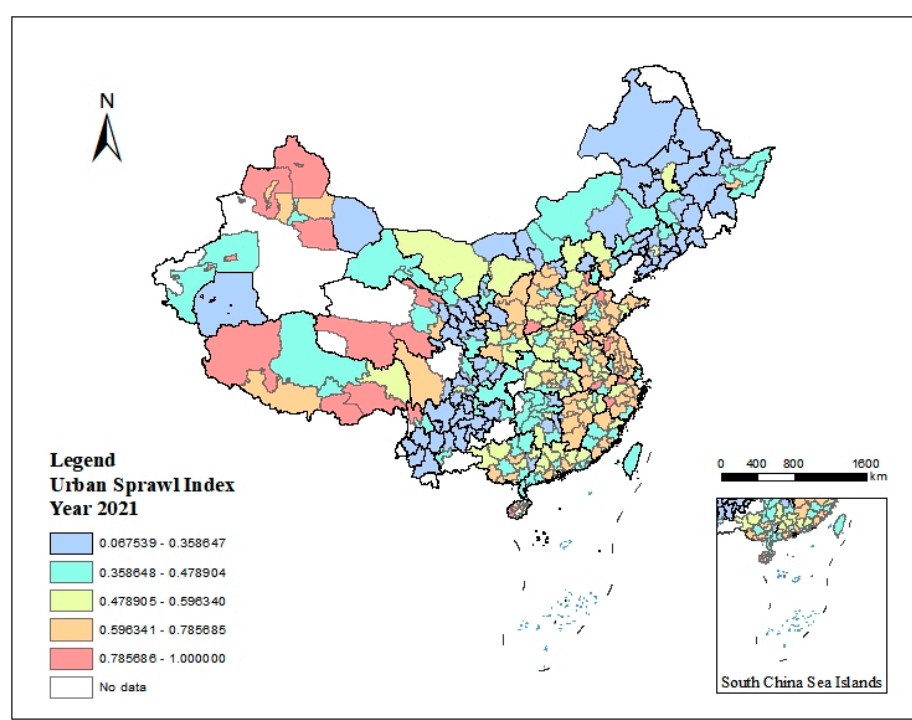

**Figure 6.** Distribution of urban sprawl by province in China, 2021. Source: LandScan global population data, 2021.

### 4.2.3. Control Variables

The regional-level control variables in this study include the natural logarithm of house price, the ratio of fiscal revenue to expenditure, GDP per capita, the share of foreign investment in GDP, the Internet penetration rate, and the urbanization level. Additionally, the enterprise-level control variables consist of enterprise age, enterprise size, gearing ratio, nature of the enterprise, shareholding ratio of the first-largest shareholder, share of fixed assets, and the growth rate of operating income. Table 1 presents the detailed definitions of these variables.

**Table 1.** Definition of variables.

| Variables | Definition |
|---|---|
| lnpatent | Logarithm of the number of patent applications in the current year plus one |
| sprawl | Continuous variable |
| sprawl$^2$ | Continuous variable |
| lnsalare | Logarithm of house prices |
| revexp | Ratio of fiscal revenues to fiscal expenditures |
| gdpave | Logarithm of per capita GDP |
| forgdp | Ratio of foreign direct investment to regional GDP |
| intpop | Internet penetration |
| urbare | Urbanization rate |
| FirmAge | ln(current year − year of incorporation + 1) |
| Size | Natural logarithm of total assets for the year |
| Lev | Total liabilities at year-end divided by total assets at year-end |
| SOE | Whether it is a state-owned enterprise: state-controlled enterprises take a value of 1; otherwise, 0 |
| Top1 | Number of shares held by the largest shareholder/total number of shares |
| Fixed | Proportion of net fixed assets to total assets |
| Growth | (current year's operating income/previous year's operating income) − 1 |

4.2.4. Regional Integration

The level of regional integration is measured by the market segmentation index; a lower index value indicates a higher degree of integration. Since the market commodity price index can only be obtained up to 2015, we construct a market segmentation index between cities using the price index method proposed by Parsley and Wei (1996), using data from the *China Urban Statistical Yearbook* from 2001 to 2016 [60]. In terms of commodity selection, this research focuses on eight categories: food, tobacco, alcohol, and supplies; clothing; housing; recreation; education and cultural goods and services; healthcare and personal goods; transportation and communication; and household equipment and supplies and maintenance services.

First, we calculate the absolute value of the relative prices for each commodity type, denoted as $\left|\Delta Q_{ijt}^k\right|$. This measure is obtained by taking the logarithmic first-order difference of the price ratio:

$$\left|\Delta Q_{ijt}^k\right| = \left|\ln(P_{it}^k/P_{jt}^k) - \ln(P_{it-1}^k/P_{jt-1}^k)\right| = \left|\ln(P_{it}^k/P_{it-1}^k) - \ln(P_{jt}^k/P_{jt-1}^k)\right|$$

where $P$ is the commodity price, $i$ and $j$ are the neighboring cities, and $t$ indicates the year.

Second, we employ the de-mean method to eliminate the price volatility stemming from the specific attributes of each commodity:

$$q_{ijt}^k = \left|\Delta Q_{ijt}^k\right| - \overline{\left|\Delta Q_{ijt}^k\right|}$$

Third, we calculate the variance in the relative price differences ($q_{ijt}^k$) for the eight commodities between adjacent cities, denoted as $Var(q_{ijt}^k)$. Subsequently, we obtain the regional market segmentation index by averaging the price variance between a particular region and its neighboring cities:

$$segment = \sum_{i \neq j} Var(q_{ijt})/n$$

Finally, we calculate the average market segmentation index for each prefecture-level city from 2001 to 2015. Based on this average, we categorize them into 'high' and 'low' groups.

*4.3. Data Sources and Description of Variables*

The data used in this study are obtained from the 2010–2020 dataset of China's A-share listed companies, LandScan Global Population Data, and the China Urban Statistics Yearbook. To ensure data quality, samples with invalid indicators from the A-share listed company data are eliminated, including the following categories: (1) companies with abnormal trading conditions labelled as ST or *ST, (2) companies in the financial and insurance sectors subject to special regulatory requirements, and (3) companies with incomplete financial or internal management data. Table 2 shows the detailed summary statistics of the variables, comprising a total of 21,882 observations.

**Table 2.** Summary statistics of the variables.

| Variables | Observations | Average Value | Standard Deviation | Minimum Value | Maximum Value |
|---|---|---|---|---|---|
| lnpatent | 21882 | 3.767 | 1.674 | 0.693 | 10.63 |
| sprawl | 21882 | 0.411 | 0.122 | 0 | 1 |
| sprawl$^2$ | 21882 | 0.184 | 0.115 | 0 | 1 |
| lnsalare | 21882 | 2.471 | 0.733 | 0.583 | 4.023 |
| revexp | 21882 | 0.734 | 0.206 | 0.0681 | 1.107 |
| gdpave | 21882 | 9.568 | 4.011 | 0.646 | 21.55 |

**Table 2.** *Cont.*

| Variables | Observations | Average Value | Standard Deviation | Minimum Value | Maximum Value |
|---|---|---|---|---|---|
| forgdp | 21882 | 4.019 | 2.792 | 0.000322 | 28.21 |
| intpop | 21882 | 45.37 | 24.43 | 1.010 | 97.78 |
| urbare | 21882 | 110.9 | 147.0 | 0.201 | 497.1 |
| FirmAge | 21882 | 2.829 | 0.365 | 0.693 | 4.143 |
| Size | 21882 | 22.09 | 1.319 | 17.81 | 28.64 |
| Lev | 21882 | 0.400 | 0.206 | 0.00752 | 1.957 |
| SOE | 21882 | 0.316 | 0.465 | 0 | 1 |
| Top1 | 21882 | 0.341 | 0.149 | 0 | 0.900 |
| Fixed | 21882 | 0.200 | 0.147 | $1.23 \times 10^{-5}$ | 0.885 |
| Growth | 21882 | 0.361 | 13.36 | −0.985 | 1878 |

## 5. Results

### *5.1. Benchmark Regression*

5.1.1. Urban Sprawl and Enterprise Innovation

Table 3 presents the results of the benchmark regression analysis. Column (1) shows the regression results considering year-, province-, and industry-fixed effects. Referring to the three-step verification method for inverted U-shaped relationships in Haans et al. (2016) [61], first, the sprawl coefficients exhibit a positive and significant relationship, while the sprawl$^2$ coefficients show a negative and significant relationship. This supports Hypothesis 1, regarding the existence of an inverted U-shaped curve. Second, the estimated slopes for the minimum and maximum values of urban sprawl are 1.6133 and −3.3563, respectively, indicating steeper curves at the two ends. The 95% confidence interval of $-\beta 1/2\beta 2$, which is [0.4080, 0.5191], falls between the minimum and maximum values of urban sprawl. Obviously, the regression results have been tested in three steps to verify Hypothesis 1 (i.e., urban sprawl exhibits an inverted U-shaped relationship with enterprise innovation). Moderate levels of urban sprawl promote enterprise innovation by concentrating factors in suburban areas, such as emerging industrial parks. However, excessive urban sprawl leads to inefficient resource allocation, hindering enterprise innovation. Columns (2) and (3) include additional control variables at the city level and enterprise level, respectively. Notably, the conclusion remains robust across these specifications.

**Table 3.** Benchmark regressions and moderating effect regressions.

| Variables | Benchmark Regressions | | | Level of Regional Integration | |
|---|---|---|---|---|---|
| | lnpatent (1) | lnpatent (2) | lnpatent (3) | Above Average (4) | Below Average (5) |
| sprawl | 1.6133 *** | 2.3825 *** | 2.1719 *** | 2.0121 *** | 1.7874 *** |
| | (0.5009) | (0.3508) | (0.5112) | (0.4861) | (0.4384) |
| sprawl$^2$ | −2.4848 *** | −2.7492 *** | −2.3427 *** | −2.0963 *** | −2.0353 *** |
| | (0.4531) | (0.3032) | (0.4653) | (0.5736) | (0.3080) |
| lnsalare | | 0.2534 *** | 0.1934 *** | 0.1189 * | 0.2749 ** |
| | | (0.0584) | (0.0468) | (0.0594) | (0.0965) |
| revexp | | 0.6723 *** | 0.6300*** | 1.0620 *** | 0.2810 |
| | | (0.0883) | (0.0859) | (0.0733) | (0.2030) |
| gdpave | | −0.0050 | −0.0127 | −0.0262 *** | −0.0127 |
| | | (0.0070) | (0.0080) | (0.0079) | (0.0142) |

**Table 3.** *Cont.*

| Variables | Benchmark Regressions | | | Level of Regional Integration | |
| | lnpatent (1) | lnpatent (2) | lnpatent (3) | Above Average (4) | Below Average (5) |
|---|---|---|---|---|---|
| forgop | | −0.0023 | 0.0010 | 0.0180 | 0.0066 |
| | | (0.0062) | (0.0038) | (0.0123) | (0.0066) |
| intpop | | 0.0038 ** | 0.0055 *** | 0.0067 *** | 0.0033 |
| | | (0.0017) | (0.0015) | (0.0019) | (0.0022) |
| urbare | | −0.0009 *** | −0.0009 *** | −0.0007 *** | −0.0010 * |
| | | (0.0002) | (0.0002) | (0.0002) | (0.0006) |
| FirmAge | | | −0.2089 *** | −0.1716 *** | −0.2262 ** |
| | | | (0.0473) | (0.0438) | (0.0864) |
| Size | | | 0.6795 *** | 0.6513 *** | 0.7137 *** |
| | | | (0.0347) | (0.0586) | (0.0219) |
| Lev | | | 0.0155 | 0.1336 | −0.0923 |
| | | | (0.1806) | (0.1594) | (0.1990) |
| SOE | | | 0.1188 *** | 0.1195 * | 0.0845 * |
| | | | (0.0273) | (0.0684) | (0.0468) |
| Top1 | | | −0.3114 *** | −0.2056 | −0.4320 ** |
| | | | (0.0781) | (0.1364) | (0.1854) |
| Fixed | | | −0.9889 *** | −0.7035 ** | −1.2473 *** |
| | | | (0.3314) | (0.3241) | (0.3268) |
| Growth | | | −0.0026 *** | −0.0026 *** | −0.0017 |
| | | | (0.0002) | (0.0002) | (0.0032) |
| Control variables | 3.5605 *** | 2.1613 *** | −11.8199 *** | −11.4892 *** | −12.2570 *** |
| | (0.1243) | (0.1851) | (0.7298) | (1.4892) | (0.5248) |
| Province-fixed effects | YES | YES | YES | YES | YES |
| Industry-fixed effects | YES | YES | YES | YES | YES |
| Year-fixed effects | YES | YES | YES | YES | YES |
| *N* | 21882 | 21882 | 21882 | 10955 | 10642 |
| $R^2$ | 0.1667 | 0.1757 | 0.4069 | 0.3971 | 0.4300 |

Note: Standard errors are reported in parentheses. ***, **, and * indicate significance at the statistical levels of 1%, 5%, and 10%, respectively.

5.1.2. Urban Sprawl, Regional Integration, and Enterprise Innovation

The movement of the inflection point is often used to determine the moderating effect of a U-shaped curve (Haans et al., 2016) [60]. We divided the samples into high and low groups based on the average level of regional integration and conducted a grouped regression on the model. As shown in columns (4) and (5) in Table 3, when the degree of regional integration is above the average, the innovation level of enterprises reaches its peak at an urban sprawl level of 0.4799. Conversely, when the degree of regional integration is below the average, this peak is reached at an urban sprawl level of 0.4391. Compared with the lower integration level, the inflection point of the U-shaped curve arrives later under a higher integration level. As shown in Figure 7, when the degree of regional integration is higher, the peak of the promotive effect of urban sprawl on enterprise innovation is reached later. Thus, Hypothesis 2 is confirmed: the degree of regional integration positively moderates the inverted U-shaped relationship between urban sprawl and enterprise innovation.

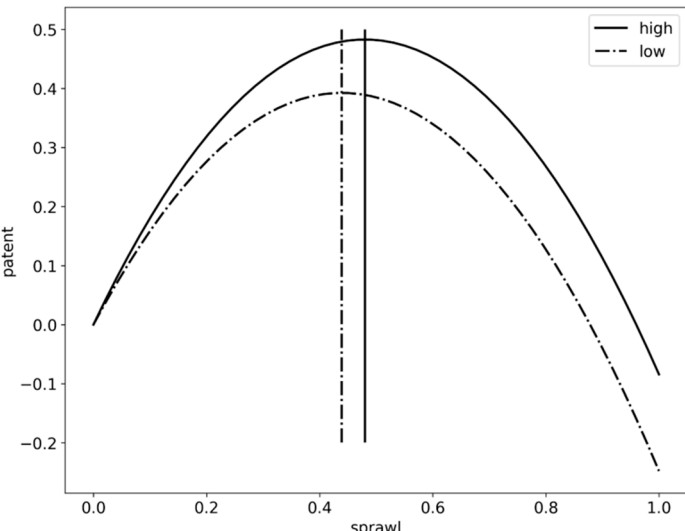

**Figure 7.** Inverted U-shaped relationship between the degree of regional integration on urban sprawl and enterprise innovation.

*5.2. Robustness Checks*

5.2.1. Two-Stage Least-Squares Regression

There is a potential issue of reverse causality between urban sprawl and enterprise innovation, which can result in biased estimates in the OLS analysis. Urban sprawl has the potential to affect enterprise innovation by influencing factors such as utilization efficiency and productivity. Conversely, more innovative and dynamic enterprises may strategically establish industrial parks or branch offices on the outskirts of cities, thereby contributing to urban sprawl. To address the endogeneity problem arising from this two-way causality, this study employs two-stage least-squares (2SLS) regression. The instrumental variables include the cross-multiplier of terrain undulation with the two-period-lagged rebar price index, the cross-multiplier of slope with the two-period-lagged aluminum price, the one-period-lagged urban sprawl index, and the one-period-lagged square of the urban sprawl index.

The selection of the cross-multiplier term between the degree of terrain undulation and the rebar price index (The data on terrain undulation are from the China Terrain Undulation Kilometer Grid dataset (You et al., 2018) [62], and the data on the rebar price index are from the Choice Financial Terminal database.) with a two-period lag is based on the following rationale. First, the degree of terrain undulation and rebar price are positively correlated with urban sprawl. Due to geographical constraints, regions with high terrain undulation tend to have lower population densities, but the levels of urban sprawl are higher. Additionally, Curci (2015) demonstrated that building height is positively associated with steel usage [12]. However, higher rebar prices lead to a reduction in steel usage, resulting in decreased building height and accelerated urban sprawl. Second, the degree of terrain relief and the rebar price index are not directly correlated with enterprise innovation.

The reasons for selecting the cross-multiplier between slope and a two-period lag of aluminum price (The average slope data are based on the digital elevation data of the ASTER Global Digital Elevation Model V003 developed by NASA, which is obtained by using ArcGIS for slope calculation. The aluminium price data are obtained from the aluminium futures settlement price of the CEEI Shanghai Futures Exchange.) are as follows. First, a higher slope is associated with lower population density, and lower population density is positively correlated with urban sprawl. When the price of aluminum rises, the cost of constructing high-rise buildings increases, leading to a decrease in high-rise buildings and a subsequent promotion of urban sprawl. Second, there is no direct correlation between slope, aluminum price, and enterprise innovation.

The results of the 2SLS regression are shown in Supplementary Material Table S1. First, according to the results of the first-stage regression, the cross-multiplier of terrain undulation and the two-period-lagged rebar price index are significantly negatively correlated with the endogenous variables (sprawl, sprawl$^2$), and the cross-multiplier of slope and the two-period-lagged aluminum price are significantly negatively correlated with the endogenous variables. The one-period-lagged urban sprawl index shows a significant positive correlation with the endogenous variables. The relationship between the squared urban sprawl index with one lagged period and sprawl is positive, though not statistically significant. The relationship between the squared urban sprawl index with one lagged period and sprawl$^2$ is significantly positive. Second, both the Cragg–Donald Wald F statistic and the Kleibergen–Paap rk Wald F statistic exceed the critical value of 7.56 at the 10% level, indicating that there is no weak instrumental variable problem. Finally, the *p*-value of the Hansen J statistic exceeds 10%, thereby confirming the hypothesis that the instrumental variables are simultaneously exogenous and affirming the absence of an over-identification problem. Based on the aforementioned analysis, the selection of instrumental variables in this study is deemed reasonable. In column (3) of Supplementary Material Table S1, the estimated coefficient of sprawl is 3.5425, while the estimated coefficient of sprawl$^2$ is −3.8829, both significant at the 1% level. This implies that, even when considering potential endogeneity, a significant inverted U-shaped relationship persists between urban sprawl and enterprises' innovation.

### 5.2.2. Regression by Year

Considering the potential variations in the effect of urban sprawl on enterprise innovation across different years, we conducted group regressions for each year. Supplementary Material Table S2 presents the results. With the exception of 2013 and 2017, the estimated coefficients of sprawl consistently show a significant positive relationship, while the estimated coefficients of sprawl$^2$ consistently demonstrate a significant negative relationship. This consistent result indicates a robust and significant inverted U-shaped relationship between urban sprawl and enterprise innovation in most years. Notably, the most pronounced inverted U-shaped relationship is observed in 2015. This finding can be attributed to the implementation of the urbanization reform program by the National Development and Reform Commission in 2014. The policy of 'removing counties and establishing districts' gained momentum during this period, leading to urban sprawl and a lagged effect of urban sprawl. As a result, the effect of urban sprawl on enterprise innovation was most pronounced in 2015.

### 5.2.3. Substituting Variables

Here, we employ research and development expenditures as a measure of enterprises' innovation, and the regression results are shown in column (1) of Supplementary Material Table S3. The estimated coefficient of sprawl is 1.9688, while that of sprawl$^2$ is −2.3049. Both coefficients are highly significant at the 1% level, aligning with the findings of the benchmark regression analysis.

Furthermore, patents include three types: inventions, utility models, and designs. For each type, we take the logarithm of the number of patent applications plus one as a proxy variable for firm innovation. Columns (2)–(4) in Supplementary Material Table S3 show the regression results for these variables, which are consistent with the findings of the benchmark regression analysis.

### 5.2.4. Other Robustness Tests

First, compared with other cities, municipalities directly under the central government enjoy unique advantages in terms of national policies and urban functions. Thus, we exclude them from the estimation sample in this subsection; as shown in Supplementary Material Table S4, the coefficients are highly significant at the 1% level, aligning with the findings of the benchmark regression.

Second, to control for the time-trend effects specific to industries and provinces and alleviate potential endogeneity issues arising from omitted variables, we control the cross-multiplication fixed effects of industry and time and of province and time. Finally, we consider the potential lagged effects of urban sprawl. We estimate each explanatory variable with one lag and two lags. The results in Supplementary Material Table S4 show that all of the aforementioned test findings remain consistent with the benchmark regression results.

## 6. Further Analysis

### 6.1. Heterogeneity of Region

Considering variations in urban development, industrial structure, and geographic conditions across regions, we divided the sample data by region into east, central, west, and north-east. The results, as presented in Supplementary Material Table S5, reveal that the coefficients of sprawl and sprawl$^2$ consistently show significant effects across all regions. Notably, the north-east region shows the most pronounced effect. This observation can be attributed to the fact that firms in the north-east region are mainly engaged in heavy industries and can effectively leverage factor resources in regions with moderate urban sprawl. Conversely, excessive urban sprawl can elevate production costs and reduce land use efficiency, thereby adversely affecting enterprise innovation.

### 6.2. Heterogeneity of City Size

Differences in economic development, infrastructure, and factor agglomeration capacity among cities can result in varied effects of urban sprawl on enterprise innovation. To address this issue, we employ group regressions based on city size (Cities with populations greater than 5 million are defined as large cities, those with populations between 2 and 5 million are defined as medium-sized cities, and those with populations below 2 million are defined as small cities). As shown in columns (1)–(3) of Supplementary Material Table S6, the estimated coefficients of sprawl and sprawl$^2$ are statistically insignificant for enterprises situated in large and medium-sized cities. In contrast, enterprises located in small cities exhibit significantly positive estimated coefficients for sprawl and negative estimated coefficients for sprawl$^2$. The probable reason is that small cities tend to be sparsely populated and more resource-rich, and moderate urban sprawl facilitates enterprises to establish sites in the expansion area and effectively utilize local factor resources. Consequently, the effect of urban sprawl on the innovation of enterprises in small cities is more pronounced.

### 6.3. Heterogeneity of Enterprise and Industry Characteristics

We conduct additional group tests to capture the diverse effects of urban sprawl on enterprises' characteristics and industries' characteristics. As shown in columns (4) and (5) of Supplementary Material Table S6, compared with manufacturing enterprises, the inverted U-shaped relationship between urban sprawl and innovation is more pronounced for non-manufacturing enterprises. The possible reason is that non-manufacturing industries are mostly service industries, which are affected by their nature and organizational management and have higher innovation dynamics, thus leading to a more significant effect of urban sprawl on innovation in non-manufacturing enterprises.

The results in columns (1) and (2) of Supplementary Material Table S7 show that the inverted U-shaped relationship between urban sprawl and innovation is more significant for enterprises above the mean in terms of size. This can be explained by the capital advantages enjoyed by larger enterprises, which facilitate innovation and contribute to the agglomeration of talent factors in the sprawl area. In columns (3) and (4) of Supplementary Material Table S7, the estimated coefficients of sprawl and sprawl$^2$ are positive and negative, respectively, passing the significance test. These findings indicate that the inverted U-shaped relationship between urban sprawl and enterprise innovation holds true for enterprises of different sizes. Notably, the inverted U-shaped relationship of urban sprawl's effect on innovation is more pronounced for enterprises with an age above the mean. This phenomenon may be attributable to the accumulation of research and development capital

and innovation experience in enterprises with an age above average, thus amplifying the significance of the inverted U-shaped relationship between urban sprawl and innovation in these enterprises.

In addition, as shown in columns (5) and (6) of Supplementary Material Table S7, the estimated coefficients of sprawl are statistically insignificant for state-owned enterprises. For non-state-owned enterprises, the estimated coefficients of sprawl and sprawl$^2$ are positive and negative, respectively, and pass the significance test. These results indicate that the inverted U-shaped relationship between urban sprawl and innovation is more pronounced for non-state-owned enterprises. One plausible explanation is that non-state-owned enterprises receive greater support and subsidies through government policies. As a result, non-state-owned enterprises are more dynamic in innovation and more motivated to effectively utilize land, labor, and resources.

## 7. Conclusions

Urban sprawl is an important phenomenon in the process of urbanization that changes the spatial structure and population distribution of cities. Investigating the influence of urban sprawl on enterprises' innovation enables a deeper understanding of the mechanisms through which urbanization affects corporate activities and sheds light on the role of urban expansion in shaping the innovation environment. In this empirical study, we investigate the effect of urban sprawl on enterprise innovation using city-level and listed company-level data from China. We employ nonlinear regression modelling and instrumental variable techniques to analyze the relationship. Several conclusions and insights are given below.

Firstly, the level of urban sprawl in Chinese cities is generally high, with heterogeneity in its regional and temporal distribution. From a temporal perspective, the level of urban sprawl in cities in the north-east has gradually declined, while in the eastern coastal regions, it has gradually increased, and in the western regions, it has consistently remained high. From a regional perspective, the level of urban sprawl is higher in the western regions and lower in the central and north-eastern regions. In response, the government should intensify efforts to implement strategies such as the development of the western regions to attract more population and mitigate the level of urban sprawl in these areas. For the eastern regions, relevant departments should strictly limit the addition of new construction land to prevent the excessive expansion of urban space, thereby alleviating the level of urban sprawl in these areas.

Secondly, there is a clear inverted U-shaped relationship between urban sprawl and corporate innovation. Specifically, moderate sprawl facilitates corporate innovation, while excessive sprawl has a negative impact on it, consistent with the findings of Brulhart and Sbergami (2009), Lin (2011), and others [33,34]. These scholars approached from the perspective of industrial agglomeration, analyzing the inverted U-shaped relationship between agglomeration and corporate productivity, thus laying the groundwork for the nonlinear discussion in this article. For areas where the urban spatial structure is too compact or excessively agglomerated, government departments should plan urban spatial structures rationally, using moderate expansion to alleviate the congestion effects of over-agglomeration and harness the scale economic effects of secondary centers to enhance corporate productivity. For areas with overly loose urban spatial structures or excessive urban expansion, city planners should strictly limit the disorderly expansion of built-up areas to avoid problems such as low resource utilization efficiency and rising transportation costs, thereby increasing research and development investments and promoting corporate innovation.

Thirdly, regional integration has played a role in moderating the relationship between urban sprawl and corporate innovation. The findings of this article confirm previous research that regional integration can impact corporate innovation (Liu et al., 2023) [20]. While Liu et al. (2023) only examined the linear effects of regional integration implementation on corporate innovation, this article discusses the moderating role of regional integration within a nonlinear research framework, representing an innovative application of regional integration [20]. Specifically, in cases of moderate urban sprawl, higher levels of regional

integration reduce barriers to the flow of knowledge, information, and other innovative elements, benefiting corporate innovation. However, under conditions of urban sprawl, higher regional integration can increase transportation costs and decrease the efficiency of resource allocation, thereby crowding out corporate research and development investments and suppressing innovation. In response, regions should promote the development of regional integration among cities, enhance the integration of public services and infrastructure between cities to fully leverage the agglomeration effects, reduce production costs for businesses, and create a favorable external economic environment for enterprises.

Finally, the relationship between urban sprawl and enterprise innovation exhibits multifaceted heterogeneity. Specifically, the inverted U-shaped effect of urban sprawl on enterprise innovation is particularly pronounced in the north-east region and in small cities. It is crucial, therefore, to develop policies that guide cities towards a polycentric model, thereby enhancing the efficient utilization of factors and fostering enterprise innovation. Additionally, at the industry and enterprise level, the influence of urban sprawl on innovation is more significant for enterprises that exceed the mean in size and age, non-state-owned enterprises, and non-manufacturing enterprises. For these enterprises, establishing manufacturing operations in urban sprawl areas may be a favorable option. Such areas tend to have less competition in factor prices, possess abundant resources, and offer a favorable environment for production activities, thereby enhancing innovation capabilities. The implementation of policies and measures tailored to different types of enterprises and regions will harness the effect of urban sprawl on enterprise innovation and promote sustainable economic development.

Unlike previous studies which primarily focused on the negative impacts of urban sprawl on corporate innovation, this paper finds that varying degrees of urban sprawl can have different effects on corporate innovation, providing a unique example of the complex impacts of urban sprawl on innovation and enriching the research in this field. Previous research has been somewhat limited on the topic of urban sprawl and corporate innovation, with most studies focusing on the dispersed employment structure in polycentric cities (Mcmillen, 2004; Cladera et al., 2009) and the promotion of innovation through compact urban forms and agglomeration (Hamidi et al., 2019; Tang et al., 2021), but lacking direct research on the impact of urban sprawl on innovation [2,3,13,17]. Urban sprawl, as a phenomenon in the urbanization process, represents a developmental direction opposite to compact and agglomerative urban forms, characterized by low-density and dispersed development in its spatial structure. Thus, studying the relationship between urban sprawl and innovation is of great practical significance. China, as one of the largest developing countries in the world, faces issues like urban sprawl in its rapid urbanization process, making it a valuable case study that can offer an example for the international community, especially developing countries, in addressing urban sprawl. This can also provide important references for other nations to mitigate the adverse effects of urban sprawl on corporate innovation and economic development. However, this paper also has certain limitations that need to be addressed in future research. Firstly, the sample selection in this study is primarily based on data from China's A-share listed companies, but there are still many private and unlisted companies in China, so the results may not be entirely accurate. We will continue to focus on this issue in future research. Secondly, this paper only examines the moderating role of regional integration, but does not investigate whether the impact of urban sprawl on corporate innovation is moderated by other factors, which will be a direction for our future research.

**Supplementary Materials:** The following supporting information can be downloaded at: https://www.mdpi.com/article/10.3390/land13050710/s1.

**Author Contributions:** Conceptualization, Z.H. and J.L.; Methodology, Z.J. and C.Y.; Investigation, Z.J.; Resources, Z.J.; Writing—original draft, Z.H.; Writing—review & editing, B.Z., C.Y. and J.L.; Supervision, C.Y.; Project administration, Z.J.; Funding acquisition, B.Z. All authors have read and agreed to the published version of the manuscript.

**Funding:** This research was funded by Major Program of National Natural Science Foundation of China (No. 42293270) and Social Science Fundation of Jiangsu (No. 22EYA001).

**Data Availability Statement:** The data presented in this study are available on request from the corresponding author. The data are not publicly available due to privacy restrictions.

**Conflicts of Interest:** The authors declare no conflicts of interest.

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
