# Peer review of "Can Urban Sprawl Promote Enterprise Innovation? Evidence from A-Share Listed Companies in China"

_land, doi:10.3390/land13050710_

Round 1

Reviewer 1 Report

Comments and Suggestions for Authors

The paper is well excecuted, nothing to say as for the case study analysis' contents. However, update the figures as it is difficult to read the legend and the text in the figures overall.

Apart from the modifications needed for the figures, the main issue detected regards the theoretical framework.

While the Authors state that "Land development in cities worldwide is typically governed by regulations imposed by local governments (Gyourko and Molloy, 2015).", I highly suggest to add the following "local governments (Gyourko and Molloy, 2015), being influenced by the intertwien with state-led policies" with a proper citation that suggest a strong influence by the state-led decisions and policies: https://www.sciencedirect.com/science/article/pii/S0264837723003770

This new text is also connected to the following phrase: "The expansion of urban areas, including urban sprawl, often arises from land transactions conducted by these local governments."

Also, in terms of urban sprawl, it would be good for the paper to insert a brief reflection on:

- spreading densities in metropolitan areas (https://onlinelibrary.wiley.com/doi/abs/10.1111/j.0022-4146.2004.00335.x)

- polycentrism (https://journals.sagepub.com/doi/10.1177/0042098009346329)

- distinction between compact and sprawl (https://journals.sagepub.com/doi/10.1080/0042098042000309748)

Please, try to add some reflections also on these themes in the concluding section. 

I am looking forward to see the revised version of this interesting paper.

Author Response

Dear Editors and Reviewers,

Thank you very much for carefully reading this manuscript. We are very grateful to your helpful and valuable recommendations for further improvements of the paper. We have revised the manuscript according to all of reviewers’ comments and suggestions. All the revised contents have been marked red in the manuscript, and the answers to the reviewers’ comments have been marked dark blue in this response letter. More details are as follows.

Reviewer 2 Report

Comments and Suggestions for Authors

The authors use statistics to investigate the effect of urban sprawl on enterprises’ innovation. The paper has some deficiencies and requires revision.

1. The introduction section is long and poorly referenced. Please provide sources/references for data and statements. For example, “… the traditional research framework assumes a linear relationship between urban sprawl and innovation”. What makes a traditional research framework in this context? Also provide references.

2. The inverted U-shaped curve is not an innovation in urban studies. Just to mention the research of Jacobson and Prakash from 1971 on urbanization and development. What is innovative about the U-shaped curve used in this article?

 3. Figure 3 and related text should be moved in the results section. In addition, please ensure that all charts and maps are clear and accurate and help to understand the content, providing detailed legends and instructions.

4. The discussion section needs improvement. An article should compare the research results with the theoretical framework, but currently, this is not the case.

Author Response

(The authors gave the same response as above.)

Reviewer 3 Report

Comments and Suggestions for Authors

Dear authors, your study seems very interesting, the method and data presented in the study provide a solid framework for analyzing the relationship between urban sprawl and enterprise innovation in China. I suggest a reflection on the selection of explanatory variables so that the reader can better understand the limitations, robustness, and validity of the results. I would like to see a discussion, as the study lacks one. Also, future lines of research within this theme seem like an interesting reference.

This study discusses urban expansion as a natural consequence of urbanization and its economic implications. It focuses on aspects of creating new areas with a demand for skilled labor, leading to the creation of new jobs. According to the article, there is an increase in business and economic power as urbanization grows.

The study is relevant as it examines urban sprawl and enterprise innovation. Methodologically, it applies robust regression analysis, which enhances credibility in the results. However, the explanatory variables used could be further explored, especially regarding their limitations.

What does it add to the subject area compared with other published
material? By specifically focusing on China, the study addresses a significant gap in the literature

What specific improvements should the authors consider regarding the
methodology? What further controls should be considered? For example, while variables such as company age, and size are common. Have the authors observed variables such as the influence of stakeholders in the innovation process or the company's market penetration, or even the internet penetration rate? I didn't quite understand if it was considered in enterprise innovation.

The conclusions are ok but...I would like to see a formal discussion of the results in this study as I believe it would be a strong addition to the research.   Regarding the figures, generally, the authors should take care to make not only the text within the figure but also its legends legible. This aspect should be improved.    

Best regards

Author Response

(The authors gave the same response as above.)

Reviewer 4 Report

Comments and Suggestions for Authors

Dear Authors,

Please answer the following review questions and revise your research paper according to them as much as needed.

  1. What type of relationship does the study identify between urban sprawl and enterprise innovation?
  2. Which types of enterprises exhibit a significant relationship with urban sprawl and innovation, according to the study's findings?
  3. In which regions and city sizes does urban sprawl have a more pronounced effect on enterprise innovation, as per the study?
  4. According to the research, how does regional integration affect the relationship between urban sprawl and enterprise innovation?
  5. What are some of the broader implications of this study for understanding the economic effects of urban sprawl on enterprise innovation?

Author Response

(The authors gave the same response as above.)

Round 2

Reviewer 1 Report

Comments and Suggestions for Authors

The paper can be accepted in the current form.